# Current Technology Developments Can Improve the Quality of Research and Level of Evidence for Rehabilitation Interventions

**DOI:** 10.3390/s23020875

**Published:** 2023-01-12

**Authors:** Bruno Bonnechère, Annick Timmermans, Sarah Michiels

**Affiliations:** 1REVAL Rehabilitation Research Center, Faculty of Rehabilitation Sciences, Hasselt University, 3590 Diepenbeek, Belgium; 2Technology-Supported and Data-Driven Rehabilitation, Data Science Institute, Hasselt University, 3590 Diepenbeek, Belgium; 3Department of Otorhinolaryngology, Antwerp University Hospital, 2650 Edegem, Belgium

**Keywords:** rehabilitation, new technology, validation, study design, methods

## Abstract

The current important limitations to the implementation of Evidence-Based Practice (EBP) in the rehabilitation field are related to the validation process of interventions. Indeed, most of the strict guidelines that have been developed for the validation of new drugs (i.e., double or triple blinded, strict control of the doses and intensity) cannot—or can only partially—be applied in rehabilitation. Well-powered, high-quality randomized controlled trials are more difficult to organize in rehabilitation (e.g., longer duration of the intervention in rehabilitation, more difficult to standardize the intervention compared to drug validation studies, limited funding since not sponsored by big pharma companies), which reduces the possibility of conducting systematic reviews and meta-analyses, as currently high levels of evidence are sparse. The current limitations of EBP in rehabilitation are presented in this narrative review, and innovative solutions are suggested, such as technology-supported rehabilitation systems, continuous assessment, pragmatic trials, rehabilitation treatment specification systems, and advanced statistical methods, to tackle the current limitations. The development and implementation of new technologies can increase the quality of research and the level of evidence supporting rehabilitation, provided some adaptations are made to our research methodology.

## 1. Introduction

For centuries the practice of medicine was based on clinical judgment and doctors’ intuition, rather than on scientific evidence. At this time, medicine was referred to as an art—“the art of Medicine”—rather than a science. In the 1960s, thanks to the development of modern research, the amount of data was growing, and some researchers, led by Alvan Feinstein, started to challenge the way medicine was performed, bringing the risk of bias into attention that could affect clinical judgment [1]. A few years later, Archie Cochrane highlighted the lack of scientific evidence supporting the practices and treatment commonly used [2]. He called for an international register of randomized controlled trials, and for explicit quality criteria for appraising published research. “Evidence-based medicine” (EBM) slowly started to emerge, and the practice was now quickly evolving. EBM was defined as “a systemic approach to analyse published research as the basis of clinical decision making” [3]. Although the awareness of the importance of basing clinical decisions on strong scientific evidence began years before, the use of EBM started in clinics in the 1990s. The development of EBM, and the modification of patients’ management, cannot be detached from the development of modern and clinical epidemiology [4]. 

The cornerstones of EBM are still—currently—randomized controlled trials (RCT), which are studies with the highest level of evidence [5]. However, these studies have some limitations: they are expensive and time-consuming, since many patients need to be included to reach enough statistical power and lower the risk of bias [6]. 

On the other hand, over the last few years, there have been significant advances and developments in modern epidemiology, both in medicine (e.g., genetics and immunology in medicine) and in rehabilitation, with the growing development of new technologies (e.g., sensor-based assessment, robotics, virtual reality, brain stimulation, etc.), allowing for objective measurements and technology-supported rehabilitation [7].

RCTs are currently not fully adapted to the development of precision therapy (i.e., personalized rehabilitation programs), or are difficult to put in place quickly in the context of emerging infectious diseases (e.g., the COVID-19 pandemic) [8]. EBM has become the benchmark for medical intervention, and today it is increasingly the benchmark for other healthcare professions. This is why we now prefer to use the term Evidence-Based Practice (EBP). For a healthcare provider, regardless of its activity, EBP is the combination of three elements [9]: (1) the provider’s own clinical expertise, (2) scientific evidence, usually in the form of practice guidelines, and (3) the preferences and values of each individual patient. However, to date, the concept of EBP has only been partially transferred to physical rehabilitation, and most rehabilitation interventions are only corroborated by a low level of evidence [10].

Therefore, the aim of this narrative review is to summarize the different study designs currently available in rehabilitation research and discuss their limitations. Then, the current limitations and future perspectives of EBP are discussed in the context of current rehabilitation interventions.

## 2. Current Situation and Limitations of the Research and Its Translation to the Care

### 2.1. Study Design and Level of Evidence

The traditional flow of development and the different validation steps of a new treatment are presented in Figure 1 (adapted from the drug development pipeline since these numbers are not available for the development of new interventions or of technology in rehabilitation) [11]. It is interesting to note the very large number of participants (healthy subjects and then patients) required to carry out the various stages of clinical validation, as well as the length of time between the development of a product and its launch on the market. The different study designs and their levels of evidence are presented in Figure 2. Some authors question this pyramid, highlighting the fact that the most significant constraint of every pyramid is that the top levels gradually build on those below them—lower-quality studies can only result in poorer-quality systematic reviews and guidelines [12]; however, the different levels of evidence on which EBP is based are generally well accepted in research and in clinical practice.

Concerning drug validation, RCTs are requested by the authorities before starting discussions on the marketing of a new molecule. In the past, a new treatment had to be more efficient (superiority trial) than a placebo (i.e., sham treatment without an active substance). However, when treatment is already available (gold-standard), it is unethical to deprive half of the participants of a study of this treatment [13]. Therefore, currently, in the presence of a gold standard, the common practice is to perform equivalence or non-inferiority trials to compare the efficacy of the new drug against this gold-standard [14]. From a statistical point of view, there are some differences, but methodologically the critical point is the randomization. The randomization allows one to get rid of some potential confounding factors, but having randomization does not mean that a study is not biased. In the next part, we will discuss the main limitations of RCTs and the sources of error in (para)medical research [15], and thus of the meta-analyses that derive from it [16].

**Figure 2 sensors-23-00875-f002:**
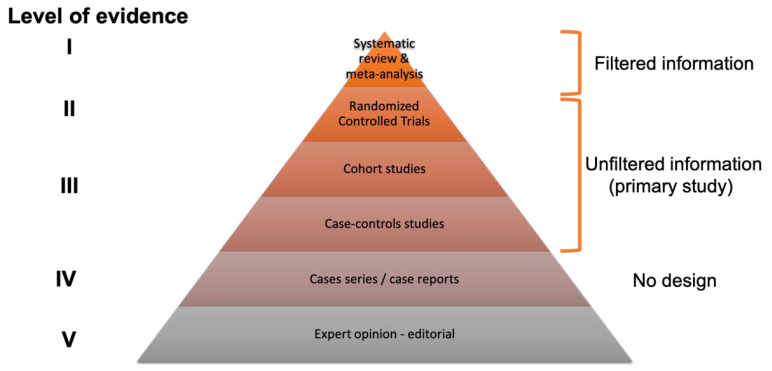
Study design and levels of evidence [16]. This pyramid could be more detailed, but the general idea is to differentiate the four kinds of studies: The meta-analysis of published studies is at the top (e.g., systematic review and meta-analysis), as level I evidence. Then the experimental studies—fully controlled (RCT) and pseudo-RCT (level II) quasi-experimental designs (i.e., prospective studies). The observational large-scale studies (cohort studies and case–control studies (level III)) and the case reports or case series are at the base of the pyramid, with a low level of evidence (level IV). Finally, expert opinion, not based on any scientific data or evidence, represents the lowest level of evidence (level V).

### 2.2. Challenges in the Validation of New Intervention and Suggestions for Reconsideration of the Choice of the Outcomes

The complexity and challenges related to the validation of a new intervention can be summarized in two constraints—time and cost, which will lead to various challenges, such as small sample sizes and low statistical power, bias, and low external validity in the studies.

Most of the time and financial resources are deployed during Phase 3 of clinical development (see Figure 1 for the different steps, phases, and their definitions). We present the different study designs according to the time and money needed to perform them in Figure 3. Because they require strict control, regular testing (e.g., biological testing, imaging, functional assessment) and many patients to have sufficient statistical power, the RCTs are—by far—the most expensive studies. As an example—again taken from the pharmaceutical world since these numbers are not known in the rehabilitation sciences—in the USA, the average cost of a Phase 1 study (healthy volunteers study) ranges from USD 1.4 million (pain and anesthesia) to USD 6.6 million (immunomodulation), including estimated site overheads and monitoring costs of the sponsoring organization. A Phase 2 study (small sample size study in patient population) costs from USD 7.0 million (cardiovascular) to USD 19.6 million (hematology), whereas a Phase 3 study (large scale clinical study) ranges in cost from USD 11.5 million (dermatology) to USD 52.9 (pain and anesthesia), on average [17]. Therefore, such studies cannot be conducted without sponsors—mainly pharmaceutical companies—and this can lead to a conflict of interest [18]. In the wake of numerous scandals that have warred the scientific community [19], clinical trial governance frameworks have been developed for pharmaceutical industry-funded clinical trials [20]. However, the recent retractions of studies focused on COVID-19 treatments have highlighted the existing relationships between researchers, private companies, and scientific journals, leading to public mistrust in the independence of research [21]. Furthermore, several studies have found a significant association between funding sources and pro-industry conclusions [22].

Another issue is the sample size needed to complete these studies. In the case of rare, or very rare, diseases, it is sometimes impossible to recruit enough patients in an RCT to reach the required statistical power [23]. It has, for example, been clearly shown that a lack of statistical power is one of the main limitations of current research in neuroscience [24], but the situation is exactly the same in the field of rehabilitation [25]. In addition to the small number of subjects included in the studies, intention to treat analysis is not always straightforward in rehabilitation (e.g., loss in follow-up, change of rehabilitation strategies according to the evolution of the patients through the rehabilitation process and its specific needs) [26]. This point may weaken the power of individual studies and RCTs. While RCTs in all (para)medical specialties are subject to loss to follow-up, rehabilitation trials have a poor track record of both reporting loss to follow-up and attaining adequate follow-up. As a result, minimizing follow-up loss should become a critical methodological priority in rehabilitation research.

RCTs are the interventional studies with the highest level of evidence. However, their results are not always reliable and unbiased [27]. Strict guidelines have been developed to increase the quality of the studies and decrease the risk of bias [28]. The protocol must be strictly followed, the allocation, the treatment, and the assessment should be performed blind, and ideally, the blind statistical analysis should be carried out by an external team. Adhering to these measures ensures that the results of the study can be trusted (internal validity). These parameters are, however, rather difficult to implement in rehabilitation research. In rehabilitation research, the blinding of the patients or the therapists is not always possible. Other problems such as low participant numbers due to the low incidence of the studied diseases, high heterogeneity, and a lack of consensus on how to demonstrate a treatment’s effect (choice of the best outcomes) are also common. Additionally, one of the specificities of rehabilitation is to propose a personalized treatment adapted as much as possible to the specificities and needs of the patient; therefore, in daily clinic the treatment is very often not fixed in time. For these reasons, treatment is often referred to as the “black box of rehabilitation” [29,30]. This lack of precise definitions, standardization and specifications of the interventions used in the studies [31] is responsible for the lack of replicability in RCTs in rehabilitation [32], and this has limited the establishment and synthesis of evidence-based practice in rehabilitation [33]. Note that efforts are being made to improve reporting, such as the establishment of CERT guidelines on exercise reporting from the equator network [34].

Furthermore, there are also some discrepancies between the results of RCTs and real-life results [35]. Two mains factors explain this (thus affecting external validity): the treatment adherence is lower in real life compared to the strict and controlled RCT environment [36], and the other issue is selection bias (or representative bias) [37]. It is well known that patients participating in clinical trials are not representative of the total patient population [38]. For example, in asthma and chronic obstructive pulmonary disease, RCTs often represent only a minority (5 to 10%) of the routine care population in whom licensed interventions will be applied [39].

A last important limitation of the research on rehabilitation, which is particularly important in the validation of new technologies, is the very important development speed of these technologies. We have seen that the validation process is time-consuming, and therefore, some validation studies may only become available when the device is already outdated, due to limited sustainability (e.g., mobile applications).

Combining the results of different studies in a meta-analysis helps to address, partially, the external validity issues (multiple clinical centers involved, different populations, different teams of clinicians, etc.), but above all it increases the statistical power by increasing the number of participants [40]. It is worth mentioning that one should distinguish between statistically significant differences (that can be achieved with the inclusion of more patients) and the clinical relevance of the observed difference (independent of the sample size) [41]. Unfortunately, although they offer the highest level of evidence, it does not mean that the conclusions of meta-analyses cannot be biased [27]. There are two main sources of bias that could influence the results of a meta-analysis. The first one is selection bias [38]: if not all the published studies are analyzed, the final results will not represent the real situation, and this is determined by the quality of the systematic review performed. The second issue is publication bias or reporting bias. This describes a phenomenon in which certain data from trials are not published, and so remain inaccessible [42]. It is well known that studies with statistically significant results have an increased likelihood of being published, and publication bias is commonly associated with inflated treatment effects, which lower the certainty of decision-makers regarding the evidence [43]. We will discuss in the next part how to deal with this phenomenon and evaluate the direction and magnitude of this bias.

## 3. New Developments Allow for Innovative Clinical Trials

In the first part, we inventoried the challenges inherent in RCTs and meta-analyses. In this section, we aim to discuss new developments and techniques that can be used to speed up research by decreasing the number of participants required, reduce the time of studies if necessary, and allow the better allocation of human and financial resources [44], while additionally reducing uncertainty.

### 3.1. Innovative Technologies (Including Sensors) to Facilitate High-Quality Clinical Trials

The development, implementation and use of technology in rehabilitation is not new. We might mention here the development of the passive system of pulley therapy to assist physiotherapists, and of course the development of continuous passive motion devices, which have revolutionized the care of patients after total knee prostheses [45]. However, over the last few decades, the technology has evolved a lot, opening up new perspectives in the management and evaluation of patients.

Rehabilitation may be aided by one or more existing technologies, such as robotics, muscle and brain stimulation, sensors-based exergames, or virtual reality [46,47].

Recent years have seen an increase in the use of robot-mediated treatment in rehabilitation to enable highly adaptable, repetitive, rigorous, and quantified physical exercise [48]. Therefore, the development of technology-supported innovative rehabilitation solutions appears to be a promising way to solve the above-mentioned limitations of current research in rehabilitation, where clinical outcomes are not sensitive enough to small changes and continuous assessment and evaluation seem impossible. The technology that is, or can be, used in rehabilitation can be divided into three categories: (i) high-tech devices the price and complexity of use of which limit their use in specialized centers (i.e., robotic treadmill); (ii) devices that can be used by clinicians in their daily practice (i.e., serious games exercises with virtual reality headsets), and (iii) systems and solutions that can be used by patients alone at home (i.e., sensors-based system, mobile health applications). 

Technology-supported rehabilitation has gained appeal due to its ability to give an objective and, if necessary, blinded assessment, which can be automated (time saver) and allows for the measurable evaluation of motor function by taking into account the characteristics of the patients [49] (e.g., kinematics, activity level, intensity, muscle activity, co-contraction, posture, smoothness of the motion during the rehabilitation exercises, heart rate, stress level, etc.), as well as their therapy adherence [48]. Today, the development of affordable and portable systems using wearables sensors is becoming more and more popular in the rehabilitation field, and such systems can, for instance, be used in daily clinics with stroke patients to assess upper limb motion [50], hand function [51], gait [52] or balance [53]. Indeed, most of the devices allow for the continuous recording of motions performed by the patients during rehabilitation exercises [54,55]. These analyses could be used later on to adapt the dose and intensity at an earlier stage of the research, or to decide to stop an intervention if it turns out that the patient will not benefit from it [56]. 

Mobile health technologies (wearable, portable, body-fixed sensors, or home-integrated devices) that measure mobility in unsupervised, everyday living situations are indeed gaining traction as forms of adjunctive clinical evaluations. Because the data collected in these ecologically valid and patient-relevant environments can capture varied and unexpected events, they may be able to overcome the limitations of routine clinical tests [57]. Remote health monitoring, based on non-invasive and wearable sensors, actuators, and modern communication and information technologies, provides an efficient and cost-effective solution that enables the patients to remain safer at home [58]. Beside the safety effect, another salient aspect of the use of new technology in rehabilitation is the remote monitoring of patients between sessions during the activities of daily living. This continuous data collection helps to detect much smaller changes in the status of the patients (i.e., improvement or deterioration) [59], and should be used in research to provide more accurate and sensitive outcomes (digital biomarkers) [60]. A digital biomarker can objectively and continuously measure and collect biological, physiological, and anatomical data through digital biosensors [61]. In addition, continuous, objective monitoring can reveal disease characteristics not observed in the clinic [62]. Different sensors can be used to perform this longitudinal data collection and assessment; for example, single- [63] or multiple-accelerometer/inertial sensor systems [64], smartwatches [65], connected insoles [66], etc., or other types of technologies (e.g., markerless camera, smartmat), can be used to assess and monitor patients at home. Due to their huge availability, another important field of development is the use of smartphone-based digital biomarkers for both assessment and to drive the rehabilitation process [67]. There are two different types of digital biomarkers: active (supervised) ones, whereby the participants have to perform specific tasks such as cognitive tasks [68], and passive (unsupervised) ones, whereby the outcomes are derived from the natural use of a smartphone, a technique known as digital phenotyping [69]. Digital phenotyping can be used to monitor age-sensitive cognitive and behavioral processes [70], but also the evolution and fluctuation of chronic diseases, such as Parkinson’s disease [71]. In this study, the authors created a single score based on a combination of supervised and unsupervised assessments (speeded tapping, gait/balance, phonation, and memory). This score was predictive of self-reported PD status and correlated with the clinical evaluation of disease severity [71].

Interestingly, this monitoring can be coupled with direct feedback and better communication with the care team, which may result in better goal setting. Studies show that continuous follow-up is not only useful for monitoring the evolution of patients and gathering data, but also has a direct effect on the progress of the patients, and leads to a significant reduction in rehospitalizations among those receiving the continuous monitoring [72]. 

Remote health assessments that collect real-world data outside of clinic settings necessitate a thorough understanding of appropriate data collection, quality evaluation, analysis, and interpretation techniques. Therefore, future development must focus on the integration of information from the rehabilitation and unsupervised assessments to better evaluate the efficacy of rehabilitation interventions.

### 3.2. Adaptative Trials

In the past, patient allocation in RCTs was done equally between the different groups, and the study went on until the end (i.e., the inclusion and follow-up of the required number of patients). This poses two major threats: it does not allow one to stop the study in the event of a higher occurrence of side effects in the treated group, or in the event of a more favorable evolution in the treated group, in which case it is unethical to continue giving non-effective treatment to half of the subjects included in the study. Adaptive trial designs permit planned data-driven modifications during the trial program, which may reduce costs, accelerate study timelines, address challenges with recruitment and heterogeneity, and mitigate inefficiencies associated with non-responders. The use of existing information to model estimated outcomes, explore statistical methods, and model the effects of adaptations to the operational elements of the trial without compromising the trial’s integrity and validity, is essential to such designs [73,74]. Interim analysis may also be used to adapt the required sample size based on the current results or stop the study, if the revised sample size is deemed to be inappropriate [75]. Finally, the adaptive trial design has been proposed as a means to increase the efficiency of RCTs [76], as it is more flexible than interim analysis. It has multiple advantages for both the patients (increasing the odds of benefiting from the treatment) and the researchers (reducing the cost and increasing the speed), while increasing the likelihood of finding a real benefit [77]. Different adaptative trials have been developed, suited to both Phase 2 (e.g., effective doses and dose-response modeling) and Phase 3 trials [78,79]. The adaptive nomenclature refers to making prospectively planned changes to the future course of an ongoing trial based on the analysis of accumulating data from the trial itself, in a fully blinded or unblinded manner, without undermining the statistical validity of the conclusions [76].

In rehabilitation, to the authors’ best knowledge, this type of study design is not yet being implemented, but it would be particularly well adapted to studies on rare pathologies (i.e., recruitment difficulties for research on acute lateral sclerosis) and/or on expensive and long treatments (i.e., exoskeleton, robotics). Some of the many factors implicated in the failure of conventional RCTs may be mitigated through the use of more adaptable, flexible methods to make trials “smarter”. For rehabilitation, given the limited research resources in some facilities and the relatively small population, it may be impossible to conduct additional studies concurrently. Therefore, reducing the length of time required to complete a study could encourage more research and the expedited development of novel methods [80].

### 3.3. Advanced Statistical Methods to Increase the Efficiency of the Research

Adaptative trials rely on the development of robust intermediate or continuous statistical analyses. New methods, such as Bayesian adaptative designs [81] and deep learning [82], are also used in development, and will help to increase follow-up and allow for the quicker modification of trial designs, in order to speed up the process and allow a maximum number of patients to benefit from the best treatment, while guaranteeing the power necessary for the study. Describing these different methods is beyond the scope of this article, but we could not discuss adaptative trials without at least mentioning the development of the statistical method. 

Indeed, changes in the design of the studies are made after analyses are carried out by the statisticians (most of the time, external companies analyze the results blindly). The development of data sciences and the implementation of new techniques in the field of rehabilitation also open up new perspectives for the agile adaptation of the treatment, ultimately leading to precision treatment [56]. However, it must be noted that for deep learning methods, and even more so for artificial intelligence techniques, a huge amount of data is required to train the systems in order to ensure high-quality results. It is not possible for clinicians to collect this type of data manually, and it is therefore essential to use new technologies (see Section 3.1), including smartphones and mHealth, to collect a large quantity of data, but also and above all to ensure the quality of these data [83].

We have previously seen that personalized treatment conflicts with the strict protocols for RCTs [29,84,85]. If the personalization uses the same algorithms for all participants, protocolization is ensured, and there is no more conflict with the RCT set up. 

Concerning the meta-analysis, the two main biases were the selection bias and the publication bias. As regards the selection bias, researchers are working on automated methods (e.g., text mining) that could improve and speed up the selection of studies, but this method is still under development [86]. On the other hand, concerning publication bias, the statistical method (i.e., trim and fill method) has been developed to estimate the size of the effect and to correct it [87]. Bayesian methods can also be used to minimize this bias [88].

## 4. Perspectives on the Development of Scientific Evidence in Rehabilitation

The goal of clinical research is to improve the health and quality of life of the patients. For this, EBP has been developed to guide practitioners in their daily practice and to ensure that they integrate the latest research optimally into their treatments. In this last part, we are going to discuss how to ease and increase the translation between research and clinical practice, focusing on pragmatic trials and the development of the rehabilitation treatment specification system.

### 4.1. Pragmatic Trials

As stated above, one of the most significant limitations of RCTs is the weak translation between research results and clinical reality [35]. The two main limitations of the translation are the treatment adherence, which is much lower in real clinical conditions compared to RCTs, and the representativeness of the population participating in the RCTs. 

Improving treatment adherence may have a more significant influence on the health of our population than the discovery of any new therapy [89]. Although the factors favoring, or, on the contrary, hindering the treatment have been well identified, it is estimated that patients are nonadherent to their treatment 50% of the time [89]. In rehabilitation, this problem occurs only in relation to the exercises that the patients need to perform at home in addition to the sessions under the supervision of the physiotherapist [90]. Solutions are being developed to increase adherence by decreasing the frequency of face-to-face sessions [91], motivational interviewing [92], and smartphone applications that support treatment instruction, track adherence, and provide patients with education and feedback on their performance [93]. To be closer to the clinical reality, some studies can be performed under real-life conditions. Although closer to reality, the problem with this type of approach is that it risks losing the well-controlled aspect that is specific to RCTs. Another limitation is that due to the larger heterogeneity in participants or interventions, larger sample sizes are often needed than in well-controlled RCTs [94]. To combine both the positive aspects of RCTs and the real-world data, some authors have suggested using a hybrid approach comprising randomization coupled with the use of pragmatic outcomes [95].

Thanks to the development of technologies, home-based assessment and monitoring are becoming popular in research and practice [96], and smart homes can be used to collect medical data that can be used for follow-up and monitoring in RCTs [97]. In addition to allowing a quick response in the case of the detection of abnormal values, this kind of approach also allows us to collect a huge amount of data that could be used in machine learning to find more sensitive outcomes.

The second limitation, the lack of representativeness of the patients included in the studies, is trickier to handle. Most of the research is performed in university hospitals, representing only a specific part of the population [98]. The problem of access to care is highly multidimensional, concerning not only financial resources but also health literacy, social support, the representation of the disease, etc. [99].

### 4.2. Rehabilitation Treatment Specification System

Although significant advances have been made in measuring the outcomes of rehabilitation interventions, comparably less progress has been made in measuring the treatment processes that lead to improved outcomes [100]. The field of rehabilitation still suffers from the black box problem: the inability to characterize treatments in a systematic fashion across diagnoses, settings, and disciplines, so as to identify and disseminate the active aspects of those treatments [101]. The rigorous definition of rehabilitation treatments, supported by theory, has been proposed in the framework of the Rehabilitation Treatment Specification System (RTSS) [100]. In the future, accurate measurements of the dose and intensity should be integrated within the RTSS to make it more closely representative of the clinical reality, and thus improve the reporting, as these parameters are crucial in rehabilitation [102].

## 5. Call for Action: The Development of Evidence-Based Technology Supported Rehabilitation

Although EBP is the current trend in healthcare practice, some clinicians and physicians are above all focused on the limitations of this approach (i.e., overall reductionism). According to them, EBP does not represent a scientific approach to health and care—it is only a restrictive interpretation of the scientific approach to clinical practice [103]. Another major limitation pointed out by some authors is the dehumanization of the patients, who are reduced to a set of numbers without taking into account social and human aspects [104]. The development and implementation of the International Classification of Functioning, Disability and Health (ICF) in rehabilitation, at least in research, is no longer in question [105]. However, while in clinics clinicians try to integrate the different components in their treatment, and to adapt the treatments from session to session, in research we are still miles away from the concept of personalized or precision rehabilitation [106]. The concept of personalized rehabilitation is closely associated with black box rehabilitation [85]. We have seen that remains a limitation in the replication of studies on rehabilitation. However, the latest and most advanced statistical methods allow for new perspectives to emerge in the management of black box rehabilitation. One of the prerequisites of this analysis is to describe as precisely as possible the different interventions and techniques used—using, for example, the RTSS framework that incorporates dose and intensity, and the outcomes derived from the ICF. The different components of the black box could then be analyzed individually to assess their specific effects. These approaches will also allow us to identify which patients will most likely benefit from a certain intervention in a certain moment of the rehabilitation process.

Currently, most physiotherapists perceive EBP as useful and necessary, but it is important to note that there is a gap between perceived and actual knowledge of EBP [107]. We think that if we improve EBP by integrating its different components, clinicians would be more likely to support it and use it in their practice. It is also important to teach clinicians about the importance of this process [108].

## 6. Conclusions

Although RCTs have been, and still are, considered the most robust form of clinical studies, we have seen that they present a risk of bias, similar to meta-analysis, and are sometimes difficult to apply in the rehabilitation field. To resolve some of the weakness of RCTs, mainly the length of study and the number of patients needed, adaptative trials have been developed and are increasingly used in medical research, but this is not yet being applied (or is to a very limited extent) in the field of rehabilitation. The development of technology-supported rehabilitation offers unique perspectives to monitor the progress of patients during the rehabilitation process. The data collected should be used to increase the quality of the trial by allowing the blinding of assessors and automated standardized data collection.

## Figures and Tables

**Figure 1 sensors-23-00875-f001:**
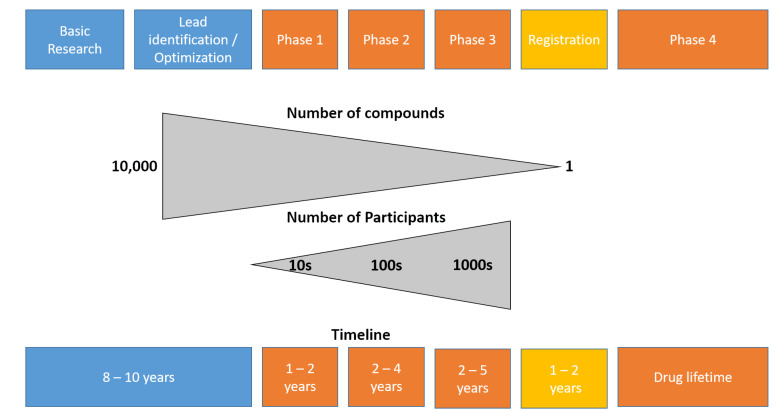
The rocket model, adapted from Verweij et al., 2019 [11]. Note that this model and its numbers relate to the development of new drugs, as no data are available for new interventions in rehabilitation sciences. Blue colors indicate the different steps of the discovery, development and preclinical research, orange represents the clinical development. The different steps of the clinical development are as follows. Phase 1—healthy volunteer study: this is the first time the drugs is tested in people; less than 100 volunteers are usually involved, and the pharmacokinetics, absorption, metabolism, and excretion effects on the body, as well as any adverse effects associated with safe dose ranges, will be determined. Phase 2—small sample size study in patient population: Evaluates the safety and effectiveness of the medicine in an additional 100–500 patients who may receive a placebo or a previously utilized standard of care. The analysis of the ideal dosage strength aids in the development of schedules, while adverse events and dangers are documented. Phase 3—large-scale clinical study: typically enrolls 1000–5000 patients, allowing medication labeling and adequate drug usage instructions. Phase 3 studies need substantial cooperation, planning, and coordination and control on the part of an Independent Ethics Committee (IEC) or an Institutional Review Board (IRB) in preparation for full-scale manufacturing after medication approval.

**Figure 3 sensors-23-00875-f003:**
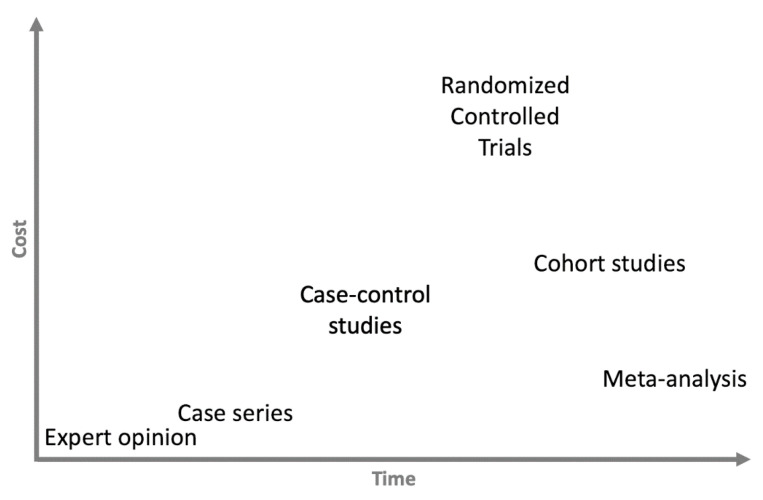
Relationship between time and cost of the different study design.

## Data Availability

Data sharing not applicable.

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
