# Peer review of "Current Technology Developments Can Improve the Quality of Research and Level of Evidence for Rehabilitation Interventions: A Narrative Review"

_sensors, 2023, doi:10.3390/s23020875_

Round 1

Reviewer 1 Report (Previous Reviewer 3)

Thank you for the opportunity to review the manuscript entitled “Current technology developments can improve the quality of research and level of evidence for rehabilitation interventions a narrative review.

This is a well written manuscript, and a very interesting approach to a real problem in research. As the authors pointed out, there are clear challenges to RCTs and meta-analyses, and the discussion is very interesting. Additionally, the development of new technologies in clinical practice has made the research field more challenging and evolving. These new technologies should be applying in research as a great data source.

The paper is clear, and I believe this narrative review apports a point of view to the readers.

Author Response

Dear reviewer,

Thank you for taking the time to review this paper and for recognizing the importance and quality of it.

Best regards,

Bruno Bonnechère

Reviewer 2 Report (Previous Reviewer 1)

Excellent review but in a number of places as indicated evidence should be cited. also, the write up requires some proof reading.

Author Response

Dear editor,

Thank you so much for taking the time to produce valuables comments and for spotting these errors in the manuscript. We really appreciated this careful reading. We have now made the changes and proofread the paper carefully with the help of a native speaker. We have now added more references to current research and publications to further support the statements as suggested.

Best regards,

Bruno Bonnechère

Reviewer 3 Report (New Reviewer)

The paper presents a review on the effectiveness of technology being used in in rehabilitation research. Starting from conventional methods, the authors have developed their presentation up to the novel technologies, such as IMUs, used in rehabilitation research. They have compared the advantages and disadvantages of these techniques.

The only concern I have is if the scope of this paper suites Sensors.

Author Response

Dear editor,

Thank you for taking the time to review this paper and for recognizing the importance and quality of it. This paper is for a special issue of the use of sensors and wearable technologies in rehabilitation, therefore we think it fits perfectly with the scope of this special issue and would be of interest for the readers.

Best regards,

Bruno

Round 2

Reviewer 2 Report (Previous Reviewer 1)

the revised version has addressed my recommendations. 

This manuscript is a resubmission of an earlier submission. The following is a list of the peer review reports and author responses from that submission.

Round 1

Reviewer 1 Report

The script is well prepared and well researched. it has a strong discussion section, but at places evidence should be cited to defend the statements made. there are also a number of grammatical issues which are highlighted in the text to note. a few sentences need rewriting for clarity. 

Author Response

Thank you for taking the time to produce valuables comments and for spotting these errors in the manuscript. We have now made the changes and proofread the paper carefully with the help of a native speaker. We have now added more references to current research and publications to further support the statements.

Reviewer 2 Report

The authors discuss the limitations of evidence-based practice in rehabilitation. They conclude that the possibility of systematic reviews and meta-analyses of the rehabilitation (methods / practices / trainings) is reduced due to the lack of well powered high quality randomized controlled trials. They suggest that other methods such as technology-supported rehabilitation systems, continuous assessment, pragmatic trials, rehabilitation treatment specification systems, and advanced statistical methods can improve the quality of research in rehabilitation.

I think that some of the authors suggestions regarding new methods are already applied in the field of rehabilitation. For example, the application of robots in the field of rehabilitation has been happening for many years. That’s the newest and most advanced technology. The continuous assessment that the authors are suggesting is also already applied in rehabilitation. There are portable devices that stroke patients can use daily for rehabilitation, for example.

Statistical methods are also applied but new methods require proper case and situation. For what sort of patients may such new statistical methods be used? There is a need for further description. A case study should be considered and be discussed to which such new statistical method can be applied.

The application of deep learning (line 258) that the authors suggest requires considerable amount of data. Off course, artificial intelligence is already applied in the field of medicine for breast cancer detection, for example. However, when it comes to the field of rehabilitation of specific group of patients, stroke patients for example, how can such an amount of data be gathered? Do the authors have any suggestion for the application of any sort of artificial intelligence method for rehabilitation? How can it be applied? Adding these information to the manuscript can make it possibly publishable.

Reviewer 3 Report

Thank you for the opportunity to review the manuscript entitled “Current technology developments can improve the quality of research and level of evidence for rehabilitation interventions”

This is a well written manuscript, and a very interesting approach to a real problem in research. As the authors pointed out, there are clear challenges to RCTs and meta-analyses, and the discussion is very interesting. Additionally, the development of new technologies in clinical practice have made the research field more challenging and evolving. These new technologies should be applying in research as a great data source.

I just wanted to add a comment:

Line 128. Please add the reference for this sentence. Reference 19 is supporting just the second sentence.

Author Response

Thank you for taking the time to read this paper and produce valuable comments and many thanks for finding this article interesting! We added the reference supporting this sentence.

Reviewer 4 Report

This is a very extensive review ragarding the use of current technology developments to improve the quality of research and level of evidence for rehabilitation interventions. Although the review is very well organized and well written I have one concern. Despite being quite lengthy it is overall quite generic. The authors mention that going into specific description of the various methods is beyong the scope of the review. 

Author Response

Thank you for taking the time to read this paper and produce valuable comments and many thanks for finding this article interesting! We added more exemples to support our opinion.

Round 2

Reviewer 2 Report

I am still not sure what the contribution of the paper is. Even though the authors have added some more description to the manuscript to address my points, I still think that the manuscript does not contain the required scientific contents for publication.

Author Response

NA (see comments to editor)